# Epidemic risk of arboviral diseases: Determining the habitats, spatial-temporal distribution, and abundance of immature *Aedes aegypti* in the Urban and Rural areas of Zanzibar, Tanzania

**Fatma Saleh**[1,2]*, **Jovin Kitau**[1,3], **Flemming Konradsen**[4], **Ayubo Kampango**[5,6], **Rahibu Abassi**[7], **Karin Linda Schiøler**[4]

**1** Department of Parasitology and Entomology, Kilimanjaro Christian Medical University College, Moshi, Tanzania, **2** Department of Allied Health Sciences, School of Health and Medical Sciences, The State University of Zanzibar, Zanzibar, Tanzania, **3** Communicable Diseases Cluster, World Health Organization Country Office, Dar es Salaam, Tanzania, **4** Global Health Section, Department of Public Health, University of Copenhagen, Copenhagen, Denmark, **5** Sector de Estudos de Vectores, Instituto Nacional de Saúde (INS), Vila de Marracuene, Província de Maputo, Mozambique, **6** Department of Zoology and Entomology, University of Pretoria, South Africa, **7** Department of Natural Sciences, School of Natural and Social Sciences, The State University of Zanzibar, Zanzibar, Tanzania

* fatma.saleh@suza.ac.tz

## Abstract

### Background

In Zanzibar, little is known about the arboviral disease vector *Aedes aegypti* in terms of abundance, spatio-temporal distribution of its larval habitats or factors associated with its proliferation. Effective control of the vector requires knowledge on ecology and habitat characteristics and is currently the only available option for reducing the risk of arboviral epidemics in the island nation of Zanzibar.

### Methodology

We conducted entomological surveys in households and surrounding compounds from February to May 2018 in the urban (Mwembemakumbi and Chumbuni) and rural (Chuini and Kama) *Shehias* (lowest government administrative unit) situated in the Urban-West region of Unguja island, Zanzibar. Larvae and pupae were collected, transported to the insectary, reared to adult, and identified to species level. Characteristics and types of water containers were also recorded on site. Generalized linear mixed models with binomial and negative binomial distributions were applied to determine factors associated with presence of *Ae. aegypti* immatures (i.e. both larvae and pupae) or pupae, alone and significant predictors of the abundance of immature *Ae. aegypti* or pupae, respectively.

### Results

The survey provided evidence of widespread presence and abundance of *Ae. aegypti* mosquitoes in both urban and rural settings of Unguja Island. Interestingly, rural setting had

**Data Availability Statement:** All relevant data are within the manuscript and its supporting information file.

**Funding:** This work was supported by Danish International Development Agency (DANIDA) through phase II of the Building Stronger Universities project, at the State University of Zanzibar, Tanzania. DFC File NO.: 14-B03-TAN. https://dfcentre.com/ The funders had no role in study design, data collection and analysis, decision to publish, or preparation of the manuscript.

**Competing interests:** The authors have declared that no competing interests exist

higher numbers of infested containers, all immatures, and pupae than urban setting. Likewise, higher House and Breteau indices were recorded in rural compared to the urban setting. There was no statistically significant difference in *Stegomyia* indices between seasons across settings. Plastics, metal containers and car tires were identified as the most productive habitats which collectively produced over 90% of all *Ae. aegypti* pupae. Water storage, sun exposure, vegetation, and organic matter were significant predictors of the abundance of immature *Ae. aegypti*.

## Conclusions

Widespread presence and abundance of *Ae. aegypti* were found in rural and urban areas of Unguja, the main island of Zanzibar. Information on productive habitats and predictors of colonization of water containers are important for the development of a routine *Aedes* surveillance system and targeted control interventions in Zanzibar and similar settings.

## Author summary

Dengue is considered the most important mosquito-borne viral disease and a global public health threat. In recent decades, large scale epidemics of dengue have occurred across sub-Saharan Africa including mainland Tanzania. *Aedes aegypti* is identified as the principal vector for dengue transmission in most affected countries. In the absence of antiviral treatment and as a dengue vaccine is not readily available; dengue prevention depends largely on vector control. As mosquitoes develop resistance towards commonly applied chemical insecticides, environmental management targeting the destruction of larval habitats is recommended. In Zanzibar, little is known about *Ae. aegypti* in terms of type, magnitude, or distribution of its larval habitats. In this study, we identified the main larval habitats of *Ae. aegypti*, their seasonal variations and factors contributing to *Ae. aegypti* abundance across urban and rural settings of Unguja Island in Zanzibar. We found widespread presence and abundance of the vector with plastic and metal containers as well as car tires identified as the most important larval habitats. Season, location of water container, water storage, sun exposure, presence of vegetation and organic matter were among the factors associated with high *Ae. aegypti* abundance. This study is the first to document widespread occurrence and distribution of *Ae. aegypti* in Zanzibar and highlights the need for the establishment of a nation-wide *Aedes* surveillance program to guide the development and monitoring of targeted, context specific vector control interventions for prevention of dengue and other arboviral epidemics. *Aedes* surveillance involves periodic inspection of households and surrounding environments for presence of larvae/pupae in water-holding containers which are then targeted for larval source reduction, as well as monitoring of adult mosquito populations.

## Introduction

The genus *Aedes* includes mosquitoes that are known vectors of several arboviral diseases of public health importance including *Aedes aegypti*. *Aedes aegypti* is widely distributed throughout the tropical and subtropical regions of the world and is the main vector of dengue,

chikungunya and yellow fever viruses in Africa [1]. Dengue is considered the most important mosquito-borne viral disease and a global public health threat [2].

Dengue virus (DENV) transmission is known to be endemic in many parts of the African region, where dengue cases or outbreaks have been reported since the 1960s [3,4]. Sub-Saharan Africa is predicted to have disproportionally high transmission intensity carrying 26% of the global burden of dengue [5]. In a recent review of reported dengue from sub-Saharan Africa, Bygbjerg et al. [6] document a notable increase in geographic scope, frequency and intensity of dengue outbreaks since the turn of the millennium. Some of these outbreaks involve large number of cases and even deaths [7].

In Tanzania, there is strong evidence for the existence of endemic dengue transmission by *Ae. aegypti* with frequent outbreaks recorded in the current decade mainly in the commercial city of Dar es Salaam [8]. Reported outbreaks have occurred in 2010, 2012, 2013, 2014, 2018, with the most recent in March of 2019 lasting for about six months [8–11]. The 2019 outbreak was the largest and most widespread with more than 6,000 confirmed cases and 13 deaths by July 2019 [11,12].

Chikungunya virus (CHIKV) infection have been reported in different regions of mainland Tanzania among febrile patients in facility-based and community-based studies, suggesting endemic transmission of the virus—most likely by *Ae. aegypti* [13–19].

To date, neither dengue nor chikungunya outbreaks have been officially reported from Zanzibar. However, the likelihood of silent and endemic circulation of DENV and CHIKV cannot be ruled out. Notably, DENV infection was confirmed by RT-PCR in 9 out of 149 (6.0%) febrile outpatients presenting at the Mnazi Mmoja main hospital in 2013 [20]. In addition, a seroprevalence study conducted at Zanzibar Blood Transfusion Services in 2011, reported anti-DENV IgG in 50.6% of tested donors [21].

Escalating dengue outbreaks in mainland Tanzania, as well as recent outbreaks of chikungunya and yellow fever in neighboring countries including Kenya, Uganda and the Democratic Republic of Congo [22], denote high vulnerability of Zanzibar to arboviral infections heightened by substantial regional travel and exchange of goods [8,23]. Moreover, novel modeling techniques using ecological niche models predict increasing risk of dengue outbreaks in coming decades for the coastal regions of Tanzania including Dar es Salaam and Zanzibar [24]. Given that there are no antiviral treatment for arboviral infection and that existing yellow fever and dengue vaccines are not readily available [25], the only tangible option for arboviral disease prevention is through management of the main vectors.

Current vector surveillance and control efforts in Zanzibar focus exclusively on *Anopheles* mosquitoes, through free distribution of long-lasting insecticide-treated bed nets (LLINs) and indoor residual spraying in high risk areas, with the aim of eliminating malaria. Both methods are hampered by the development of insecticide resistance [26], while LLINs are ineffective against diurnal *Aedes* mosquitoes. Nonetheless, these efforts have contributed to a dramatic reduction in malaria prevalence and incidence in Zanzibar [26–28]. Whereas malaria cases are declining, ongoing waste generation, unreliable domestic water supply creating a demand for localized water storage, and environmental and climatic changes seem to be favoring the proliferation of *Aedes* mosquitoes [24,29]. This development underscores the need in Zanzibar for multi-disease programs involving integrated vector management [30] of both Anopheline and Aedine disease vectors, including targeting of larval habitats.

Evidence of *Aedes* larval habitats and their seasonal distribution is limited for Zanzibar. Recently, we reported findings of the first systematic survey of *Aedes* larvae/pupae on the island, which documented presence of *Ae. aegypti* mosquitoes in Zanzibar city [29]. Here, domestic water storage containers and discarded objects were identified as the most important larval habitats. In addition, presence of vegetation, organic substances and shorter duration of

sun exposure were found to be significantly associated with colonization of water containers [29].

This small-scale baseline study was, however, confined to the city center, which is environmentally (vegetation coverage, demography, housing architecture and planning) unique from the rest of the urban district and might therefore present with distinct types of habitats. Moreover, arboviral disease transmission and outbreaks may not be confined to the city center given considerable daily mobility of people and goods between urban and rural settings of Zanzibar.

In order to inform future control efforts, the objective of this study was to identify the main larval habitats, their seasonal variations and predictors of *Ae. aegypti* abundance across urban and rural settings in Zanzibar.

## Materials and methods

### Ethics statement

The study protocol was approved by the Research and Ethics Review Committee of Kilimanjaro Christian Medical University College Certificate No. 2226 dated 24[th] October 2017. The study was also approved by Zanzibar Medical Research Ethics Committee of the Ministry of Health Zanzibar and a research permit was obtained from Zanzibar Research Council in the Second Vice President's Office. All heads of households went through a standardized consent process in which the purpose and procedures of the research were explained in detail in local language (Kiswahili). The right not to participate or to withdraw at any point was also explained as was the assurance of confidential handling of all data. A written consent form was signed by all participants.

### Study area

This study was carried out on Unguja, one of two main islands in the archipelago of Zanzibar, situated off the coast of Tanzania. Zanzibar has a tropical climate with weather comprising two dry and two rainy seasons. The dry seasons span from June to September and January to February and the rainy seasons last from October to December and March to May. During the study period (February to May 2018), Unguja island had mean maximum and minimum daily temperatures between 32.0˚C and 25.0˚C in the dry season (February to early March), and 29.5˚C and 24.3˚C in the rainy season (April to May 2018), respectively. The mean relative humidity was 79.5% in the dry and 86.0% in the rainy seasons. The recorded total rainfall in the dry season was 49 mm in the urban sites (Mwembemakumbi and Chumbuni) and 30 mm in the rural sites (Chuini and Kama). The total rainfall in the rainy season was 1,139 mm and 1,196 mm in the urban and rural settings, respectively (Director Zanzibar office, Tanzania Meteorological Authority, personal communication).

The study was conducted in the Urban-West Region of Unguja consisting of two districts: the Urban District and the West District. (Fig 1). At the time of the 2012 census [31], the region had a population of almost 600,000, a density of 2,581 persons per km$^2$ and an average annual population growth rate of 4.2%. The Urban-West region has both urban and rural environments with 112,716 households in total, of which 19,320 are in rural and 93,396 are in urban areas [31]. The region has a total of 84 *Shehias* (lowest government administrative unit); 45 in Urban district and 39 in West district. The population of Urban and West districts are 223,033 and 370,645, respectively [31].

### Sampling sites

This cross-sectional entomological survey involved two study settings, in each district (Urban and West) purposely selected based on ecological and demographic characteristics. The

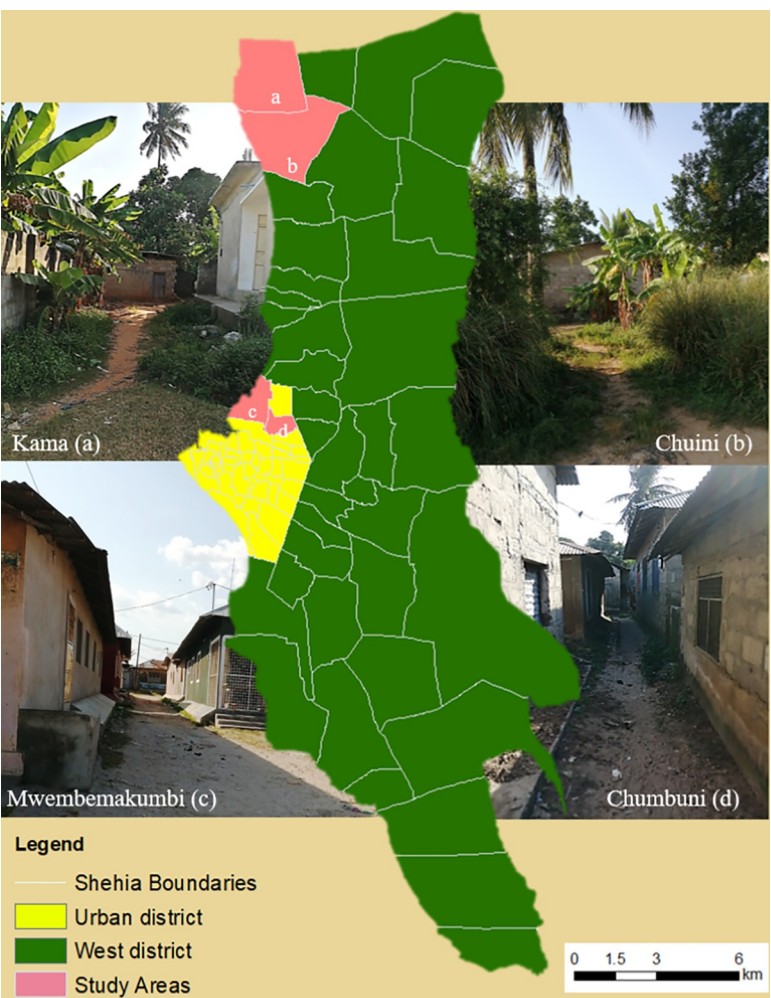

**Fig 1. Map of Urban-West region, Zanzibar displaying the geographical locations of the studied settings.** The attached photos illustrate the typical environment of each study site.

urban setting included two *Shehias*; Mwembemakumbi (6°08′54.15″S 39°12′40.68.00″E) and Chumbuni (6°08′39.19″S 39°13′04.58″E) located at the outskirts of Urban district about 2.5 km from Zanzibar Stone Town (Fig 1).

The rural setting included two *Shehias*; Chuini (6°05′37.69″S 39°14′10.02″E) and Kama (6°02′05.91″S 39°12′25.02″E) situated at the western border of West district. The *Shehia* populations are 8,354 (Mwembemakumbi), 10,925 (Chumbuni), 6,158 (Chuini) and 2,921 (Kama) [31].

### Larvae/pupae surveys

Entomological surveys were conducted from February to May 2018 by two teams of three trained field assistants. Each team carefully screened the study areas for water-holding containers. The screening included indoor and outdoor premises of households and other buildings as well as public open spaces.

The recommended sample sizes for detection of > 2% infestation in households [32] were used and increased by 10% to account for inaccessible households (refused entry or consent unavailable during data collection) [32]. Included houses in each season were randomly

selected using house IDs from a sampling frame of all households in the respective *Shehia* provided by the local community leader (*Sheha*). In addition, all public buildings such as mosques and schools in the study areas were screened.

In this study, a sample was defined as an actual or potential habitat in the form of any uncovered or partially covered container holding stagnant water [33]. Water containers harboring at least one larva or pupa were considered positive, whereas water containers with no larva or pupa were considered negative. Productivity denoted the abundance of *Ae. aegypti* pupae in a given water-holding container [34]. The characteristics of all positive and negative containers were examined and recorded on site. The containers were categorized by setting, season, location, container function, type/material of container, water volume, sun exposure, presence or absence of organic matter and vegetation (Table 1). Absence or presence of *Ae. aegypti* was recorded along with the number of larvae and pupae for each positive container identified. House index (HI = number of houses positives for *Ae. aegypti* immatures/Number of houses inspected*100) and Breteau index (BI = number of positive containers per 100 houses inspected) were calculated to estimate *Ae. aegypti* densities in the respective study settings [33]. In addition, pupae-per-person index (PPI = number of pupae/total population of the inspected households) was also calculated [35].

**Table 1. Description of variables as used in the study.**

| Variable | Category | Definition |
|---|---|---|
| **Independent** | | |
| Setting | Rural | Population density $< 1{,}200$ persons/km$^2$ |
| | Urban | Population density $> 4{,}000$ persons/km$^2$ |
| Season | Wet | April to May 2018 |
| | Dry | February to early March 2018 |
| Location | Indoor | Water containers found inside a roofed building (intra-domestic space) |
| | Outdoor | Water containers found in open areas including peri-domestic spaces |
| Function | Water storage | Water stored for specific purposes including domestic activities, pet use, gardening, construction, and religious purposes |
| | Discarded objects | Discarded water-holding containers |
| | Other water receptacles | Any water-holding container different to that of water storage and discarded objects including tree holes, leaf axils and ground pools |
| Water volume | Low volume | $<5$L of water |
| | Medium volume | 5L–20L of water |
| | High volume | $>20$L of water |
| Organic matter | Yes | Presence of organic materials in the water container such as decaying wood or leaves |
| | No | Absence of visible organic materials in the water |
| Sun exposure | Exposed half a day or less | The habitat is exposed to sun for half of the day or less |
| | Exposed more than half a day | The habitat is exposed to sun for more than half of the day |
| Vegetation | Yes | Presence of vegetation in the habitat such as algae, floating and emerged plants |
| | No | Absence of vegetation |
| **Dependent** | | |
| Presence or absence of immature *Ae. aegypti* in a given water container | | |
| Presence or absence of *Ae. aegypti* pupae in a given water container | | |
| The number of immature *Ae. aegypti* in a given water container | | |
| The number of *Ae. aegypti* pupae in a given water container | | |

### Larvae and pupae collection

Samples of mosquito larvae and pupae were collected weekly during dry (February to first week of March) and wet (April to May) seasons of 2018. In each study area, two 1-week collection periods (one in each season) were conducted. Samples were collected 3–5 days per week from 8:00 AM to 3:00 PM. Larvae and pupae from small sized containers were collected using 5ml large-mouthed plastic pipettes. Small sized positive containers with less than 20L of water were completely emptied and strained, where possible. In large containers with more than 20L of water, a 500ml plastic dipper was used for larvae and pupae collection. The immature stages were sampled using standard operating procedures for *Ae. aegypti* [34]. The collected larvae and pupae were placed in ID labeled vials with loose screw caps and kept in cool boxes for transportation to the insectary at the State University of Zanzibar.

### Rearing and species identification

Rearing of immature stages and species identification were conducted as previously described by Saleh et al. [29]. Briefly, at the insectary, the collected immatures were sorted into genera, *Aedes* larvae were separated from pupae and counted, then both were reared to adult stage for species identification. Almost 72% (N = 9,796/13,671) of the reared *Aedes* immatures emerged as adults. *Ae. aegypti* were identified under a dissecting microscope using the morphological identification key by Huang [36].

### Data analysis

Data were analyzed using R software version 4.0.2. [37]. Statistical analyses were conducted for presence or absence as well as abundance of *Ae. aegypti* for a given water container. The analyses were conducted for all immatures (i.e. both larvae and pupae) and pupae, separately. Generalized linear mixed models (GLMMs) were applied to determine the effect of the following environmental parameters (predictors): setting, season, location, function, water volume, sun exposure and presence of organic matter and vegetation (Table 1) on the presence or absence and abundance of *Ae. aegypti* immatures or pupae.

For binary outcomes (presence or absence of immatures or pupae), GLMMs with binomial distribution and *logit* link function to predictors were applied to determine the likelihood of infestation by *Ae. aegypti* immature stages, whereas negative binomial distribution with log link function were applied to determine the effect of predictors on the abundance of *Ae. aegypti* immatures and pupae, and to account for over-dispersion in both immatures and pupae counts. To determine whether adding random factor was justified, two models were fitted: one with fixed effects only, and one with mixed effects (fixed plus random factors). The best model fit was assessed by the values of Akaike Information Criterion (AIC), and the model with the smallest AIC was chosen. The day or week of study were modelled as observation level random factors to account for dependence between repeated observations made across settings over time as well as to model over-dispersion in binary outcomes [38].

In addition, Chi-square test ($\chi^2$) was applied to determine the association between environmental parameters and immature *Ae. aegypti* infestation, and the *Stegomyia* indices (HI, BI, PPI) across seasons and settings. All statistical analyses were performed at a 0.05 significance level.

## Results

### Distribution of inspected households and water containers

A total of 1,314 houses were inspected during the study period in the rural (N = 683) and urban settings (N = 631). Nearly equal numbers of houses were inspected in the

**Table 2. *Stegomyia* indices for *Ae. aegypti* by season in rural and urban areas of Zanzibar.**

| Setting | Season | Houses | | | Containers | | | | | |
| | | Inspected | Positive | HI(%) | Inspected | Positive | BI | No. of pupae | No. of persons | PPI |
|---|---|---|---|---|---|---|---|---|---|---|
| Rural | Wet | 340 | 144 | 42.4 | 565 | 359 | 105.6 | 709 | 847 | 0.8 |
| | Dry | 343 | 82 | 23.9 | 164 | 112 | 32.6 | 586 | 712 | 0.8 |
| Urban | Wet | 315 | 102 | 32.4 | 249 | 184 | 58.4 | 424 | 839 | 0.5 |
| | Dry | 316 | 46 | 14.6 | 115 | 53 | 16.8 | 198 | 638 | 0.3 |
| Total | | 1314 | 374 | 28.5 | 1093 | 708 | 53.9 | 1917 | 3036 | 0.6 |

HI = House index, BI = Breteau index, PPI = Pupae-per-person index

wet and dry seasons and in the rural (wet 340; dry 343) and urban (wet 315; dry 316) settings.

Of the 1,314 houses inspected, 374 (28.5%) were positive for immature *Ae. aegypti*. Rural setting had significantly higher number of positive houses (N = 226, 33.1%) as compared with urban setting (N = 148, 23.5%) ($\chi^2$ = 14.95, p < 0.001). Nearly twice the number of positive houses were found in the wet (N = 246, 37.6%) compared with the dry (N = 128, 19.4%) seasons. The difference was highly significant ($\chi^2$ = 53.05, p < 0.001). Higher house indices were observed in the rural, 42.4% (wet) and 23.9% (dry) than urban, 32.4% (wet) and 14.6% (dry), settings (Table 2).

During the study period, a total of 1,093 water containers were inspected, of which 729 (66.7%) and 364 (33.3%) were identified in the rural and urban settings, respectively (Table 2). The vast majority of water containers; 814 (74.5%) were identified in the wet season and only 279 (25.5%) during the dry season. Nearly equal numbers of containers were identified in indoor (N = 569, 52.0%) and outdoor (N = 524, 48.0%) spaces. As for container function, water storage constituted the highest number of containers (N = 542, 49.6%) followed by discarded items (N = 488, 44.6%) and other water receptacles (N = 63, 5.8%) (Tables 3 and 4).

A number of positive houses had two or more positive water containers in the rural (N = 104/226, 46.0%) and urban (N = 32/148, 21.6%) settings, and in the wet (N = 114/246, 46.3%) and dry (22/128, 17.2%) seasons. Of the 1,093 inspected water containers, 708 (64.8%)

**Table 3. Rural and urban distribution of water containers inspected (N) and positive for *Ae. aegypti* and number of *Ae. aegypti* immatures[a] by season, location, and function of water container in Zanzibar.**

| Parameter | Category | Rural | | | Urban | | |
| | | No. of containers | | | No. of containers | | |
| | | N (%[b]) | Positive (%[c]) | No. of immatures (%[b]) | N (%[b]) | Positive (%[c]) | No. of immatures (%[b]) |
|---|---|---|---|---|---|---|---|
| Season | Wet | 565 (77.5) | 359 (63.5) | 5,795 (67.1) | 249 (68.4) | 184 (73.9) | 3,379 (67.2) |
| | Dry | 164 (22.5) | 112 (68.3) | 2,847 (32.9) | 115 (31.6) | 53 (46.1) | 1,650 (32.8) |
| Location | Indoor | 295 (40.5) | 201 (68.1) | 3,318 (38.4) | 274 (75.3) | 192 (70.1) | 4,275 (85.0) |
| | Outdoor | 434 (59.5) | 270 (62.2) | 5,324 (61.6) | 90 (24.7) | 45 (50.0) | 754 (15.0) |
| Function | Water storage | 313 (43.0) | 214 (68.4) | 4,435 (51.3) | 229 (62.9) | 156 (68.1) | 3,609 (71.8) |
| | Discarded items | 383 (52.5) | 248 (64.8) | 3,981 (46.1) | 105 (28.9) | 77 (73.3) | 1,348 (26.8) |
| | Other water receptacles | 33 (4.5) | 9 (27.3) | 226 (2.6) | 30 (8.2) | 4 (13.3) | 72 (1.4) |
| Overall (%) | | 729 (100) | 471 (64.6) | 8,642 (100) | 364 (100) | 237 (65.0) | 5,029 (100) |

[a] Larvae plus pupae.

[b] Percent of water containers or *Ae. aegypti* immatures for each category within a given parameter.

[c] Percent of *Ae. aegypti* positive containers within each category.

**Table 4. Rural and urban distribution of identified water containers (N), containers positive for *Ae. aegypti* and number of *Ae. aegypti* pupae by season, location, and function of water container in Zanzibar.**

| Parameter | Category | Rural | | | Urban | | |
|---|---|---|---|---|---|---|---|
| | | No. of containers | | | No. of containers | | |
| | | N (%[a]) | Positive (%[b]) | No. of pupae (%[a]) | N (%[a]) | Positive (%[b]) | No. of pupae (%[a]) |
| Season | Wet | 565 (77.5) | 139 (24.6) | 709 (54.7) | 249 (68.4) | 97 (39.0) | 424 (68.2) |
| | Dry | 164 (22.5) | 56 (34.1) | 586 (45.3) | 115 (31.6) | 26 (22.6) | 198 (31.8) |
| Location | Indoor | 295 (40.5) | 77 (26.1) | 460 (35.5) | 274 (75.3) | 97 (35.4) | 545 (87.6) |
| | Outdoor | 434 (59.5) | 118 (27.2) | 835 (64.5) | 90 (24.7) | 26 (28.9) | 77 (12.4) |
| Function | Water storage | 313 (43.0) | 88 (28.1) | 624 (48.2) | 229 (62.9) | 78 (34.1) | 475 (76.4) |
| | Discarded items | 383 (52.5) | 105 (27.4) | 656 (50.6) | 105 (28.9) | 43 (41.0) | 130 (20.9) |
| | Other water receptacles | 33 (4.5) | 2 (6.1) | 15 (1.2) | 30 (8.2) | 2 (6.7) | 17 (2.7) |
| Overall (%) | | 729 (100) | 195 (26.7) | 1,295 (100) | 364 (100) | 123 (33.8) | 622 (100) |

[a] Percent of water containers or *Ae. aegypti* pupae for each category within a given parameter.

[b] Percent of *Ae. aegypti* positive containers within each category.

and 318 (29.1%) were positive for *Ae. aegypti* immatures and pupae, respectively. The rates of positive containers for all immatures (N = 471/729, 64.6%) and pupae (N = 195/729, 26.7%) in the rural setting were nearly equal to that of the urban setting (immatures 237/364, 65.0% and pupae 123/364, 33.8%). Higher number of containers were positive for immature *Ae. aegypti* in the wet (N = 543/814, 66.7%) compared with dry (N = 165/279, 59.1%) seasons (Table 3). The difference was statistically significant ($\chi^2$ = 5.22, p = 0.02). Furthermore, 236 (29.0%) and 82 (29.4%) containers were positive for *Ae. aegypti* pupae in the wet and dry seasons, respectively (Table 4). However, the difference was not statistically significant (p = 0.90). The corresponding Breteau indices (BI) in the rural setting (wet 105.6; dry 32.6) were nearly double that of the urban setting (wet 58.4; dry 16.8) for both wet and dry seasons. While the PPI ranged between 0.3 and 0.8, depending on season and setting (Table 2). Nevertheless, the *Stegomyia* indices were not statistically significantly different between seasons across settings; HI ($\chi^2$ = 0.87, p = 0.34), BI ($\chi^2$ = 0.03, p = 0.87), and PPI ($\chi^2$ = 0.03, p = 0.99).

The number, types and distribution of all immature mosquitoes collected is presented in S1 Table.

## Distribution of water containers and *Ae. aegypti* by environmental parameters in the rural and urban settings

A total of 729 water containers were identified in the rural setting (Chuini and Kama) with more than three times the number of containers recorded in the wet (N = 565, 77.5%) than dry (N = 164, 22.5%) seasons. More containers were identified outdoor (N = 434, 59.5%) than indoor (N = 295, 40.5%). However, the rates of positive containers in the outdoor (N = 270, 62.2%) and indoor (N = 201, 68.1%) areas in the rural setting were not significantly different (p = 0.10). In terms of function, discarded items accounted for highest number of containers (N = 383, 52.5%), compared to water storage (N = 313, 42.9%) and other water receptacles (N = 33, 4.5%) (Tables 3 and 4). Nevertheless, no significant difference was found in the rates of immatures positive containers between discarded/other water receptacles (N = 257/416, 61.8%) and water storage (N = 214/313, 68.4%) (p = 0.07) in the rural setting. Likewise, pupae positive containers were not significantly different between discarded/other water receptacles (N = 107, 25.7%) and water storage (N = 88, 28.1%) (p = 0.47).

Of the 729 identified water containers in the rural setting, 471 (64.6%) were positive for immature *Ae. aegypti*, harboring a total of 8,642 immatures (Table 3) of which 1,295 were pupae identified in 195 (26.7%) containers (Table 4).

In the urban setting (Mwembemakumbi and Chumbuni), 364 water containers were identified, of which 249 (68.4%) and 115 (31.6%) were recorded in the wet and dry seasons, respectively. Unlike the rural setting, the number of indoor containers (N = 274, 75.3%), and rates of positive containers (N = 192, 70.1%) in the urban setting were larger than that of outdoor containers (N = 90, 24.7%, positive 45, 50.0%). The difference in indoor and outdoor positive containers were statistically significant ($\chi^2$ = 12.01, p = 0.001). Likewise, water storage containers constituted the highest number (N = 229, 62.9%), followed by discarded items (N = 105, 28.9%) and other water receptacles (N = 30, 8.2%) (Table 3). Nevertheless, the rates of immatures positive containers between discarded/other water receptacles (N = 81/135, 60.0%) and water storage (N = 156/229, 68.1%) in the urban setting were not significantly different (p = 0.12). Likewise, no significant difference was found in pupae positive containers between discarded/other water receptacles (N = 45, 33.3%) and water storage (N = 78, 34.1%) (p = 0.47) (Table 4).

Of the 364 identified water containers in the urban setting, 237 (65.1%) were positive for immature *Ae. aegypti*, harboring a total of 5,029 immatures (Table 3) including 123 (33.8%) pupae positive containers harboring 622 pupae (Table 4). The distribution of water containers and *Ae. aegypti* by other parameters (water volume, sun exposure, vegetation, and organic matter) is summarized in Table 5.

## Distribution of water containers and *Ae. aegypti* by type of container material

Plastic containers comprised nearly half (N = 529, 48.4%) of all water containers, followed by metal (N = 235, 21.5%), tires (N = 164, 15.0%), natural habitats (N = 53, 4.8%), drains and

**Table 5. Characteristics of water containers (N) inspected for immature[1] *Ae. aegypti* in Zanzibar.**

| Parameter | Category | N (%[2]) | Containers positive for immatures (%[3]) | No. of immatures (%[2]) | Containers positive for pupae (%[3]) | No. of pupae (%[2]) |
|---|---|---|---|---|---|---|
| Water volume | <5L | 642 (58.7) | 430 (67.0)[a] | 8,067 (59.0) | 190 (29.6)[a] | 1,137 (59.3) |
| | 5L–20L | 275 (25.2) | 168 (61.1)[a] | 3,415 (25.0) | 80 (29.1)[a] | 452 (23.6) |
| | >20L | 176 (16.1) | 110 (62.5)[a] | 2,189 (16.0) | 48 (27.3)[a] | 328 (17.1) |
| Sun exposure | Exposed ≤½ day | 821 (75.1) | 554 (67.5)[a] | 10,948 (80.1) | 254 (30.9)[a] | 1,579 (82.4) |
| | Exposed > ½ day | 272 (24.9) | 154 (56.6)[b] | 2,723 (19.9) | 64 (23.5)[b] | 338 (17.6) |
| Vegetation | Yes | 334 (30.6) | 265 (79.3)[a] | 5,676 (41.5) | 127 (38.0)[a] | 788 (41.1) |
| | No | 759 (69.4) | 443 (58.4)[b] | 7,995 (58.5) | 191 (25.2)[b] | 1,129 (58.9) |
| Organic matter | Yes | 433 (39.6) | 331 (76.4)[a] | 6,586 (48.2) | 156 (36.0)[a] | 882 (46.0) |
| | No | 660 (60.4) | 377 (57.1)[b] | 7,082 (51.8) | 152 (23.0)[b] | 1,035 (54.0) |
| Overall (%) | | 1,093 (100) | 708 (64.8) | 13,671 (100) | 318 (29.1) | 1,917 (100) |

[1] Larvae plus pupae.

[2] Percent of water containers or *Ae. aegypti* immatures for each category within a given parameter.

[3] Percent of water containers positive for *Ae. aegypti* immatures within each category.

The categories followed by different letters (a, b) within a given parameter were significantly different ($\chi^2$, p < 0.05) and those indicated by the same letter (a, a, or b, b) were not significantly different.

**Table 6. Distribution and abundance of *Ae. aegypti* immatures[a] and pupae by type or material of water container in Zanzibar.**

|  | All containers | %[b] | Positive containers (%[c]) | %[d] | Total immatures[a] | %[e] | Pupae | %[f] |
|---|---|---|---|---|---|---|---|---|
| Plastic | 529 | 48.4 | 344 (65.0) | 48.6 | 7,030 | 51.4 | 1,002 | 52.3 |
| Metal | 235 | 21.5 | 172 (73.2) | 24.3 | 3,599 | 26.3 | 548 | 28.6 |
| Tires | 164 | 15.0 | 134 (81.7) | 18.9 | 2,065 | 15.1 | 281 | 14.6 |
| Natural habitats | 53 | 4.8 | 22 (41.5) | 3.1 | 436 | 3.2 | 28 | 1.5 |
| Concrete Tanks | 38 | 3.5 | 14 (36.8) | 2.0 | 279 | 2.0 | 35 | 1.8 |
| Drains/pools | 50 | 4.6 | 7 (14.0) | 1.0 | 67 | 0.5 | 9 | 0.5 |
| Others | 24 | 2.2 | 15 (62.5) | 2.1 | 195 | 1.5 | 14 | 0.7 |
| Total | 1,093 | 100 | 708 | 100 | 13,671 | 100 | 1,917 | 100 |

[a]Larvae plus pupae.

[b]Percent of all containers.

[c]Percent of that type of container positive for *Ae. aegypti*.

[d]Percent of positive containers.

[e]Percent of all immatures.

[f]Percent of all pupae

pools (N = 50, 4.6%), concrete tanks (N = 38, 3.5%) and other containers such as glass bottles, ceramic and pottery (N = 24, 2.2%) (Table 6).

Together plastic, metal containers and tires contributed more than 90% of *Ae. aegypti* positive containers as well as number of all immatures or pupae alone (Table 6). Occurrence and abundance of container type by setting and season are shown in Figs 2 and 3.

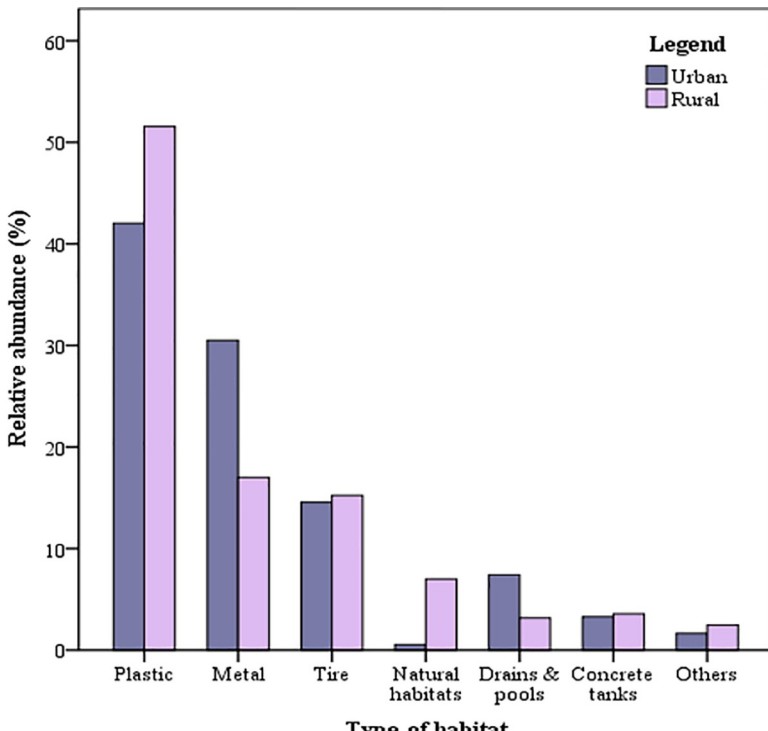

**Fig 2. Occurrence and abundance of immature *Ae.** aegypti* habitats/container type in the rural and urban settings in Zanzibar.

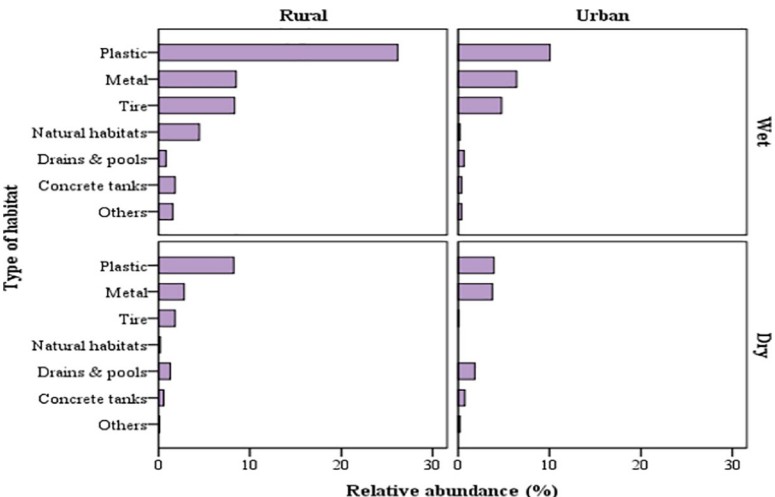

**Fig 3. Occurrence and abundance of immature *Ae.* aegypti habitats/container type by setting and season in Zanzibar.**

## Characteristics of water containers associated with *Ae. aegypti* immatures, and pupae

Of the seven environmental parameters examined, wet season (OR [Odds ratio] = 2.4, 95% CI = 1.6–3.7), indoor located containers (OR = 2.1, 95% CI = 1.4–3.1), water storage containers (OR = 1.5, 95% CI = 1.1–2.1), presence of vegetation (algae) (OR = 4.0, 95% CI = 2.8–5.7) and organic matter (OR = 2.8, 95% CI = 2.0–3.8) were all positively significantly associated with the presence of *Ae. aegypti* immatures (Table 7).

Additionally, water storage (IRR = 1.5, 95% CI = 1.2–1.9; p = 0.001), presence of vegetation (IRR = 1.6, 95% CI = 1.2–2.1; p < 0.001), and organic matter (IRR = 1.6, 95% CI = 1.2–2.0; p < 0.001) were significant predictors of the abundance of *Ae. aegypti* immatures (Table 8).

For the pupae stage specifically, water containers exposed to sunlight less than half a day (p = 0.04), and containers with vegetation (OR = 1.9, 95% CI = 1.4–2.7), and organic matter (OR = 1.7, 95% CI = 1.3–2.3) were more likely to contain *Ae. aegypti* pupae compared to those exposed to sunlight for longer durations and those without vegetation or organic matter. Furthermore, only water containers exposed to sunlight less than half a day (p = 0.01) were significant predictors of the abundance of *Ae. aegypti* pupae. The effect of other parameters was not significantly different (Tables 7 and 8).

**Table 7. Binomial models for the association between environmental parameters and *Ae. aegypti* immatures (larvae plus pupae) and pupae.**

| Parameter (category)* | Presence of all immatures | | Presence of pupae | |
|---|---|---|---|---|
| | Odds ratio (95% CI) | p-value | Odds ratio (95% CI) | p-value |
| Intercept | 0.1 (0.1–0.3) | <0.001 | 0.1 (0.1–0.3) | <0.001 |
| Setting (rural vs. urban) | 1.5 (0.9–2.4) | 0.09 | 0.9 (0.6–1.3) | 0.54 |
| Season (wet vs. dry) | 2.4 (1.6–3.7) | <0.001 | 1.3 (0.8–1.9) | 0.26 |
| Location (indoor vs. outdoor) | 2.1 (1.4–3.1) | <0.001 | 1.1 (0.7–1.6) | 0.69 |
| Function (water storage vs. discarded/others) | 1.5 (1.1–2.1) | 0.009 | 1.2 (0.9–1.7) | 0.24 |
| Sun exposure (exposed ≤½ day vs. exposed > ½ day) | 1.3 (0.9–2.0) | 0.13 | 1.5 (1.2–2.3) | 0.04 |
| Vegetation (yes vs. no) | 4.0 (2.8–5.7) | <0.001 | 1.9 (1.4–2.7) | <0.001 |
| Organic matter (yes vs. no) | 2.8 (2.0–3.8) | <0.001 | 1.7 (1.3–2.3) | 0.001 |

*Likelihood that the first category had more positive containers for immatures or pupae than the second (reference) for a given parameter

**Table 8. Negative binomial models for the association between environmental parameters and abundance of *Ae. aegypti* immatures (larvae plus pupae) and pupae.**

| Parameter (category)* | Abundance of all immatures | | Abundance of pupae | |
|---|---|---|---|---|
| | IRR[a] (95% CI) | p-value | IRR[a] (95% CI) | p-value |
| Intercept | 2.5 (3.3–9.2) | <0.001 | 1.0 (0.4–2.5) | 0.95 |
| Setting (rural vs. urban) | 1.1 (0.8–1.5) | 0.54 | 1.1 (0.7–1.7) | 0.77 |
| Season (wet vs. dry) | 0.9 (0.7–1.2) | 0.61 | 0.6 (0.4–1.0) | 0.06 |
| Location (indoor vs. outdoor) | 1.0 (0.7–1.3) | 0.85 | 0.7 (0.4–1.1) | 0.09 |
| Function (water storage vs. discarded/others) | 1.5 (1.2–1.9) | 0.001 | 1.5 (1.0–2.2) | 0.05 |
| Sun exposure (exposed ≤½ day vs. exposed > ½ day) | 1.3 (1.0–1.8) | 0.07 | 1.8 (1.2–3.0) | 0.01 |
| Vegetation (yes vs. no) | 1.6 (1.2–2.1) | <0.001 | 1.4 (1.0–2.1) | 0.07 |
| Organic matter (yes vs. no) | 1.6 (1.2–2.0) | <0.001 | 1.3 (0.9–1.9) | 0.13 |

*Likelihood that the first listed category produced more immatures or pupae than the second (reference) within a given parameter

[a]IRR = Incidence rate ratio (IRR)

## Discussion

This cross-sectional study provides the first evidence of widespread *Ae. aegypti* infestation in urban and rural settings in Unguja island, while also identifying the most favourable conditions for colonization and proliferation of this species. High vector densities were found throughout the study sites. We found HI and BI between 15% and 42% and between 17 and 106 for HI and BI, respectively, depending on season and setting. Furthermore, PPI ranged between 0.3 and 0.8, depending on season and setting (Table 2). Studies in other African countries have reported similarly high *Stegomyia* indices [39–41]. According to the WHO [33], a HI and BI of greater than 5% and 20, respectively, for any locality indicate the locality is sensitive to arboviral diseases and should be prioritized for control measures. Together with our previous findings from the city of Zanzibar [29], this study suggests high risk of arboviral disease transmission by *Ae. aegypti* throughout the island underscoring the urgent need for a nation-wide *Aedes* surveillance and control program.

Notably, we found the number of immatures (larvae and pupae) and pupae *Ae. aegypti* to be nearly twice as high in the rural compared to urban setting. In accordance with this, we found higher number of houses and water containers positive for *Ae. aegypti* in the rural compared to urban setting, irrespective of season (Table 2). Although studies in other geographical regions have identified *Ae. aegypti* as a predominantly urban species [13,42,43], it is increasingly reported in different rural and peri-urban settings [40,44,45], and recurrently associated with arboviral transmission in rural populations [19,46–51]. Moreover, a shift towards increased rural transmission as well as 'travelling waves' phenomenon in dengue occurrence have previously been reported in several dengue endemic countries [52–55] underscoring the need for well adapted vector control operations in both rural and urban settings.

The island nation of Zanzibar is notable for its small geographical scale and focal location of all major markets and services in the urban district. This generates substantial daily mobility of people and goods between the urban center and most rural communities, none of which are further away than a few hours' drive. We suspect that *Ae. aegypti* has exploited this rural-urban dynamic to spread across the island of Unguja, most likely through egg or larvae infested goods [32].

In addition, constant scarcity of piped water supply compels residents to store water for longer durations thereby creating *Ae. aegypti* larval habitats in and around premises. This coupled with poor solid waste management provide favorable conditions for proliferation of the mosquito on the island. Notably, in the rural areas we identified a large number of informal dumps with no formal system of waste collection in place. Here, discarded items accounted for the highest number of water containers compared with the urban setting where waste collection has been recently outsourced to local community groups.

Consistent with our previous study [29], a variety of water-holding containers were identified as potential habitats in both rural and urban settings, the vast majority of which were man-made. Three types of materials (plastic, metal, and car tires) constituted almost 85% of all containers and more than 90% of containers positive for *Ae. aegypti*. Plastic containers contributed more than half of the immature and pupae stages (51.4% and 53.2%, respectively), followed by metal containers (26.3% immature and 28.6% pupae stages) and tires (19.0% immature and 14.6% of pupae stages) (Table 6). The premier position of plastic containers as *Ae. aegypti* habitat is widely recognized [9,29,56–58], and could well reflect the sheer scale by which plastic containers are readily available in the pantropical environment especially in the form of water storage tanks and unmanaged waste. Further investigation is needed to ascertain the influence of container material in attracting oviposition and development of immature stages.

Interestingly, recent studies conducted in the Dar es Salaam region identified discarded items including used car tires as main habitat for immature *Ae. aegypti* [9,59,60]. In our study, however, water storage containers were shown to be significant predictors of the abundance of immature *Ae. aegypti* as compared with discarded items. Similar reports have been made by Nguyen et al. [44], Ngugi et at. [45], Lin et al. [42] and, Saleh et al. [29]. A study by Wolf-Peter Schmidt et al [51] reported a high vector/host ratio with subsequent high risk of dengue in the rural low-density areas in Vietnam caused by lack of piped water supply which compelled residents to store water thereby creating *Ae. aegypti* larval habitats. Water storage practice is also common in Zanzibar in both rural and urban areas, due to widespread scarcity of piped water and regular rationing of the supply. We identified stable *Ae. aegypti* habitats both inside and outside inspected premises in the form of water storage containers, which were kept for domestic, religious, pets/livestock, farming and/or construction purposes. The most common indoor and outdoor water storage containers included plastic tanks, steel/metal tanks, buckets, jerrycans, drums, basins/bowls, of different sizes.

Although tires presented the third most important habitat in this study producing 14.6% of pupae, it should be noted that trade with used, imported tire is common in Zanzibar. The trade in used tires is important as it presents an excellent means of transporting *Aedes* eggs to new locations [61,62]. Also, as used tires are short lived and therefore frequently replaced compared with new ones, special consideration should be placed on their storage and disposal to prevent them from becoming *Ae. aegypti* larval habitats.

As expected, we identified significantly higher numbers of immature *Ae. aegypti* in containers with vegetation (mostly algae) and organic matter as compared to those without. The importance of these biotic factors in attracting oviposition and providing nutrient sources for *Aedes* and other mosquito larvae has previously been reported, as have the protective effects of vegetation against predators [29,59,63–65]. As water storage containers are likely to retain water for a longer duration, they are likely to grow algae which serve as food for the developing larvae.

In this study, we found higher numbers of pupae in the wet than dry seasons. However, the average number of pupae per positive container was higher in dry season compared to wet season. This finding might be attributed to the fact that during dry season water is stored for

much longer durations than in the wet season when opportunities for replenishing water storage containers are more frequent. This suggests that during dry season there might be a relatively small number of stable water containers particularly those stored indoors or under shaded areas that produce large amount of *Ae. aegypti*. These containers may extend the risk of arboviral disease transmission into the dry season, as reported in other geographical settings [14,18,53,66,67].

Our study identified a small number of *Anopheles gambiae* species in the rural area of Kama. Despite substantial efforts and resources currently accorded to the elimination of malaria across the archipelago [26], parasite reservoirs are known to remain. The dual risk of malaria and arboviral diseases as uncovered by this study, underscores the need for an integrated vector management approach. Integrating *Aedes* control with the well-established Zanzibar Malaria Elimination Program (ZAMEP) could optimize the use of resources and help increase preparedness for new disease epidemics [68]. This approach is consistent with integrated vector management guidelines recommended by WHO [68].

Importantly, our findings also highlight an urgent need for the integration of water supply and waste management services into the disease control efforts, through strengthening of inter- and intra-sectoral action and collaboration [68]. For instance, the widespread dependency on single-use water bottles is considered a direct consequence of inadequate supply and quality of portable water. Empty plastic bottles, in particular, are omnipresent in both urban and rural environments, as an unmanaged waste product. A similar situation is recognized in many other sub-Saharan countries with Kenya taking the unprecedented step to ban single use plastic, including water bottles, from protected areas as of June 2020 [69]. Our study suggests that Zanzibar would be prudent to follow this or similar approaches, while improving access to safe portable water and the disposal of additional waste materials from the environment. Donor supported infrastructure programs to improve water, sanitation and waste management are reportedly at different stages of development in Zanzibar [70], yet in addition to these structural measures, it is evident that lasting behavior change must be secured at individual, household and community level if *Aedes* habitats are to be eliminated from the domestic and peri-domestic environment. Moreover, almost half (49%) of the population of Zanzibar is under 18 years [71], suggesting that sustained social and behavior change communication targeting children may present the most cost-effective approach.

We acknowledge certain limitations to our study including the possible inter-observer bias between research assistants, even as all assistants were thoroughly trained prior commencement of field surveys and swapped study sites each sampling week.

As we used purposive sampling to select study sites, our findings may not be representative across *Shehias*. However, as this is the first investigation in these localities, we consider it appropriate to conduct an exploratory study with a focus on areas more likely to harbor immature *Aedes* habitats. Notably, a simple random sampling method was used in both study settings to eliminate potential selection bias at household level.

The roof and other elevated areas of each building were not inspected in this study, with a risk that potential habitats were overlooked, especially during the wet season. However, due to the nature of housing designs and construction (mostly corrugated iron roofs without gutters) in both study sites, these kinds of habitats were considered few if not unlikely.

In addition, only visible containers with stagnant water and those with openings or partial covers were included in this study, thus omitting concealed and/or covered, yet positive, containers. Nevertheless, the containers included in this study represent the types of water containers commonly found in Zanzibar.

Finally, we acknowledge the short duration of the study including short durations of field mosquito sampling during both wet and dry seasons.

The current study provides a systematic insight to the distribution and abundance of arboviral disease vector (*Ae. aegypti* mosquitoes) in the rural and urban areas of Zanzibar and environmental factors associated with their larval habitats. The results of this study, to a large extent, support the earlier publication on *Ae. aegypti* preferred and productive habitats in Zanzibar City [29]. Malaria vectors in Zanzibar have developed resistance towards the most common chemical insecticides following persistent exposure [26]. A similar challenge is likely to occur for the control of *Aedes* unless efforts are undertaken to reduce the dependency on chemical insecticides. This requires that we understand the ecological determinants of vector distribution and habitat productivity when designing more targeted and cost-effective *Aedes* vector-control initiatives [34]. Information on productive habitats will inform the prospective *Aedes* surveillance program and subsequent development of context-specific vector control interventions including community mobilization, environmental manipulations, and modification, as well as waste and water management. As both HI and BI indices were much higher in the rural than urban setting, control interventions should not be limited to the urban district but to rural areas as well. To maximize resources and tools for vector control [30], it is recommended that a comprehensive appraisal and needs assessment of the current vector control program [72] be conducted and that *Aedes* mosquito control be integrated with the existing ZAMEP to limit the risk of arboviral epidemics. Finally, to complement these vector control efforts, the existing integrated disease surveillance and response system must be assessed and strengthened in terms of overall performance and capacity for early detection of arboviral disease transmission.

## Supporting information

**S1 Table. Distribution of mosquito genera/species by season in rural and urban areas of Zanzibar.** [a]Percent of all mosquitoes by setting or season.
(PDF)

## Acknowledgments

We are grateful to Arabia Makame, Said Suleiman, Suleiman Juma, Ali Omar, Bakar Khamis, Salum Kimbau, Moza Rashid and Mohammed Abdulrahman for their field and laboratory assistance, and Mr Abbas Mzee for creating the map for us. We thank community leaders and household owners for their cooperation during the entire duration of study, which made it successful.

## Author Contributions

**Conceptualization:** Fatma Saleh, Jovin Kitau, Flemming Konradsen, Karin Linda Schiøler.

**Data curation:** Fatma Saleh, Flemming Konradsen.

**Formal analysis:** Fatma Saleh, Ayubo Kampango, Rahibu Abassi.

**Funding acquisition:** Flemming Konradsen, Karin Linda Schiøler.

**Investigation:** Fatma Saleh.

**Methodology:** Fatma Saleh, Jovin Kitau, Flemming Konradsen, Karin Linda Schiøler.

**Project administration:** Fatma Saleh.

**Resources:** Flemming Konradsen, Karin Linda Schiøler.

**Software:** Fatma Saleh, Ayubo Kampango, Rahibu Abassi.

**Supervision:** Jovin Kitau, Flemming Konradsen, Karin Linda Schiøler.

**Validation:** Fatma Saleh.

**Visualization:** Fatma Saleh.

**Writing – original draft:** Fatma Saleh.

**Writing – review & editing:** Fatma Saleh, Jovin Kitau, Flemming Konradsen, Ayubo Kampango, Karin Linda Schiøler.

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
