## [Decision Letter · Decision Letter 0]

2 Jul 2020

Dear SALEH,

Thank you very much for submitting your manuscript "Epidemic risk of arboviral diseases: determining the habitats, spatial-temporal distribution and abundance of immature Aedes aegypti in the Urban and Rural areas of Zanzibar, Tanzania" for consideration at PLOS Neglected Tropical Diseases. As with all papers reviewed by the journal, your manuscript was reviewed by members of the editorial board and by several independent reviewers. In light of the reviews (below this email), we would like to invite the resubmission of a significantly-revised version that takes into account the reviewers' comments. Please ensure that each comment/suggestion is addressed.

In addition to the reviewers’ comments, please address my concerns below: 

1. Lines 25, 58, 59, 60, 63 and others: Although commonly used, mosquitoes do NOT breed in water. Breeding has a reproductive/sexual connotation. Mosquito larvae (and pupae) develop in water. I would use “larval habitats” in place of “breeding habitats” throughout the manuscript.

2. Line 95: main hospital in 2013 (Ali Mohammed, unpublished data). Who is Ali Mohammed? Do you have permission to site his work, where does he work, why is he not cited in the acknowledgements?

3. Lines 108-110: Bed nets tend to protect people at night, but Ae. aegypti is a day-biting mosquito.

4. Lines 140-144: Is the rainfall amount really accurate to 0.1 mm? I would tend to round all of these off to the nearest whole mm. When someone claims a degree of accuracy not supported by the data, readers tend to question all of the data.

5. Lines 144-145 (and Figure 1): This may be a little confusing. Here it says that, “…the urban and West districts that make up the Urban-West region of Unguja.” For those people not familiar with the Mjini and Magharibi districts within the Mjini Magharibi Region, they may not realize that the Urban-West Region is made up of two districts, the Urban District and the West District. Unfortunately, looking at Fig. 1 does not help as one cannot see which parts of the Region are in which district. It might be clearer if the two districts were shown in different “patterns” and the study areas highlighted.

6. Lines 146-149: Minor, but these data are from the 2012 census, i.e., several years before the actual study. Again, is it really important that 593,678 people lived there 6 years before the study, or would it be better to say that according to a 2012 census [29], the population was about “594,000” or “nearly 600,000?

7. Line 175: Minor, but “water holding containers…” should be “water-holding containers…”

8. Line 197: Minor, but “two one-week…” should be “two 1-week……” Similarly, it should be “3 to 5 days” or 3-5 days” on the next line. While numbers less than 10 are nearly always written out, for units of measurement, the numeral is used. 

9. Line 223 (and elsewhere): Minor, but when I first read this, I thought that you meant number positive for larvae and then for pupae. However, on re-reading it, I realized that you meant all immature stage (i.e., both larvae and pupae) and then just for pupae. Would it help to change “immature stages” to “all immature stages” or possibly to change larval “immature stages” to “immature stages (i.e., both larvae and pupae)” the first time this is used? Note, I noticed that the definition of immature stages was given in Tables 3, 5, and 6.

10. Lines 232-236: I am having a problem with these numbers. If 28.5% of all of the houses were positive for Ae. aegypti, how can the positivity rates be 60.4 and 39.6% for the rural and urban areas. I understand that 226 of the houses in the rural area were positive, but according to the previous sentence and the table, 683 houses were tested. Thus, shouldn’t the positivity rate for the rural area been 33.1%? The same comment applies to all of the other rates mentioned in this section.

11. Lines 247-248: Again, I am having a problem with these numbers. If 64.8% of the inspected water containers were positive, how could only 28.5% of the houses be positive? I am guessing that many of the houses actually had no water containers at all and that others had a lot of positive ones. You might want to clarify this.

12. Lines 249-250: Looking at Table 3, there appears to be 729 containers in the Rural area and 364 in the Urban area. Why were there 708 containers examined for all immature stages in both the Urban and Rural areas? Why is the denominator for pupae different than that for all immatures? To me, this sentence is saying that in rural areas, 471 of 708 containers were positive for immatures, yet according to Table 3, 471 of 729 containers were positive for immatures in the rural area. Please check all these numbers.

13. Table 5: I know that the statistics are shown in Table 8, but it might be nice to indicate the statistically significant differences. For each of the parameters (volume, sun, vegetation, and organic), you could follow each of the percent positives with a letter. If they were followed by the same letter, then they would not be significantly different, but if followed by different letters, they would be significantly different. For example, for Organic matter, Yes, would be followed by an “a” and No would be followed by a “b.” For Sun exposure, both < half day and >half day would be followed by a “a,” and for Water volume, <5L would be followed by an “a,” 5L-20L followed by a “b,” and >20L followed by a “b.”

14. Line 321: Minor, but I don’t think that “Anopheles” has been established prior to this line.

15. Line 336: In this section the authors present numerous statistics and confidence intervals for differences. However, numerous comparisons were made. Some of the CI barely don’t include 1 (e.g., line 345, for rural versus urban, the lower limit is 1.11). Is it possible that if these numbers were adjusted for multiple comparisons, that the CI would have included 1?

16. Line 345 (and elsewhere): Yes, it may be 1.53 times higher odds, but is the hundredth place really meaningful? Consider rounding all of these to the nearest 10th. See also line 351. Is the thousands place at all meaningful here?

17. Line 358-360 (and other places): The authors compared <5L and >20L multiple times, but never mentioned <5L vs 5-20L or 5-20L vs >20L. Based on Table 5, volumes of <5L should be significantly higher than both 5-20L and >20L, and 5-20L should not be significantly different than >20L. Is that correct? 

18. Tables 8 and 9: The tables merely give the p-value for whether or not the two parameters are significantly different. Although it is explained in the text, the reader can’t tell from the tables which of the two is more likely to have immatures or pupae. Yes, it does appear that you are comparing the first listed against the second listed, but it never states that. Might it be worth a footnote to “Parameter” along the lines of “Likelihood that the first parameter produced more immatures or pupae than the second parameter.” Also, what about the comparison between <5L and 5-10L. 

19. Lines 416-417: Minor, but the number of immature Ae. aegypti stages if both larvae and pupae were present would be 2. The number of “immatures” is what you are examining. I must admit that this is a minor and the readers will understand, but technically, the answer is 0, 1, or 2 for the number of stages present.

20. Line 422: Is it worth a short sentence here. We found HI and BI between 15 and 42 and between17 and 106 for HI and BI, respectively, depending on season and location.

21. Lines 452-455: Yes, I know that plastic containers contributed the most. However, was it because plastic containers were more attractive or a better habitat, or was it because there were so many more plastic containers. If you look at it slightly differently, percent of type of container that was positive for Ae. aegypti immatures, then the order is reversed, with plastic having only 65%, metal being 73%, and tires have a 82% positive rate for Ae. aegypti. This is almost a contradiction of lines 458-462.

22. Line 464 (also 478 and 536): Again, mosquitoes do not “breed” in tires, their immatures develop there.

23. Line 478: Also, the trade in used tires is important as that is an excellent means of transporting Aedes eggs to a new location.

24. Lines 498-499 and 550: why establish “ZAMEP” if you do not use it again. Either do not establish or use on line 55

25. Line 552: Why establish “IDSR” if it is not used. Please check to see if ALL of the abbreviations established are actually used. If not, then please remove these unnecessary abbreviations.

26. References: Please ensure that these are formatted according to PLoS NTD guidelines. Only the first word or proper nouns in a title should be capitalized. See reference 3 and others.

We cannot make any decision about publication until we have seen the revised manuscript and your response to the reviewers' comments. Your revised manuscript is also likely to be sent to reviewers for further evaluation.

Sincerely,

Michael J Turell, Ph.D.

Associate Editor

Amy Morrison

Deputy Editor

In addition to the reviewers’ comments, please address my concerns below: 

1. Lines 25, 58, 59, 60, 63 and others: Although commonly used, mosquitoes do NOT breed in water. Breeding has a reproductive/sexual connotation. Mosquito larvae (and pupae) develop in water. I would use “larval habitats” in place of “breeding habitats” throughout the manuscript.

2. Line 95: main hospital in 2013 (Ali Mohammed, unpublished data). Who is Ali Mohammed? Do you have permission to site his work, where does he work, why is he not cited in the acknowledgements?

3. Lines 108-110: Bed nets tend to protect people at night, but Ae. aegypti is a day-biting mosquito.

4. Lines 140-144: Is the rainfall amount really accurate to 0.1 mm? I would tend to round all of these off to the nearest whole mm. When someone claims a degree of accuracy not supported by the data, readers tend to question all of the data.

5. Lines 144-145 (and Figure 1): This may be a little confusing. Here it says that, “…the urban and West districts that make up the Urban-West region of Unguja.” For those people not familiar with the Mjini and Magharibi districts within the Mjini Magharibi Region, they may not realize that the Urban-West Region is made up of two districts, the Urban District and the West District. Unfortunately, looking at Fig. 1 does not help as one cannot see which parts of the Region are in which district. It might be clearer if the two districts were shown in different “patterns” and the study areas highlighted.

6. Lines 146-149: Minor, but these data are from the 2012 census, i.e., several years before the actual study. Again, is it really important that 593,678 people lived there 6 years before the study, or would it be better to say that according to a 2012 census [29], the population was about “594,000” or “nearly 600,000?

7. Line 175: Minor, but “water holding containers…” should be “water-holding containers…”

8. Line 197: Minor, but “two one-week…” should be “two 1-week……” Similarly, it should be “3 to 5 days” or 3-5 days” on the next line. While numbers less than 10 are nearly always written out, for units of measurement, the numeral is used. 

9. Line 223 (and elsewhere): Minor, but when I first read this, I thought that you meant number positive for larvae and then for pupae. However, on re-reading it, I realized that you meant all immature stage (i.e., both larvae and pupae) and then just for pupae. Would it help to change “immature stages” to “all immature stages” or possibly to change larval “immature stages” to “immature stages (i.e., both larvae and pupae)” the first time this is used? Note, I noticed that the definition of immature stages was given in Tables 3, 5, and 6.

10. Lines 232-236: I am having a problem with these numbers. If 28.5% of all of the houses were positive for Ae. aegypti, how can the positivity rates be 60.4 and 39.6% for the rural and urban areas. I understand that 226 of the houses in the rural area were positive, but according to the previous sentence and the table, 683 houses were tested. Thus, shouldn’t the positivity rate for the rural area been 33.1%? The same comment applies to all of the other rates mentioned in this section.

11. Lines 247-248: Again, I am having a problem with these numbers. If 64.8% of the inspected water containers were positive, how could only 28.5% of the houses be positive? I am guessing that many of the houses actually had no water containers at all and that others had a lot of positive ones. You might want to clarify this.

12. Lines 249-250: Looking at Table 3, there appears to be 729 containers in the Rural area and 364 in the Urban area. Why were there 708 containers examined for all immature stages in both the Urban and Rural areas? Why is the denominator for pupae different than that for all immatures? To me, this sentence is saying that in rural areas, 471 of 708 containers were positive for immatures, yet according to Table 3, 471 of 729 containers were positive for immatures in the rural area. Please check all these numbers.

13. Table 5: I know that the statistics are shown in Table 8, but it might be nice to indicate the statistically significant differences. For each of the parameters (volume, sun, vegetation, and organic), you could follow each of the percent positives with a letter. If they were followed by the same letter, then they would not be significantly different, but if followed by different letters, they would be significantly different. For example, for Organic matter, Yes, would be followed by an “a” and No would be followed by a “b.” For Sun exposure, both < half day and >half day would be followed by a “a,” and for Water volume, <5L would be followed by an “a,” 5L-20L followed by a “b,” and >20L followed by a “b.”

14. Line 321: Minor, but I don’t think that “Anopheles” has been established prior to this line.

15. Line 336: In this section the authors present numerous statistics and confidence intervals for differences. However, numerous comparisons were made. Some of the CI barely don’t include 1 (e.g., line 345, for rural versus urban, the lower limit is 1.11). Is it possible that if these numbers were adjusted for multiple comparisons, that the CI would have included 1?

16. Line 345 (and elsewhere): Yes, it may be 1.53 times higher odds, but is the hundredth place really meaningful? Consider rounding all of these to the nearest 10th. See also line 351. Is the thousands place at all meaningful here?

17. Line 358-360 (and other places): The authors compared <5L and >20L multiple times, but never mentioned <5L vs 5-20L or 5-20L vs >20L. Based on Table 5, volumes of <5L should be significantly higher than both 5-20L and >20L, and 5-20L should not be significantly different than >20L. Is that correct? 

18. Tables 8 and 9: The tables merely give the p-value for whether or not the two parameters are significantly different. Although it is explained in the text, the reader can’t tell from the tables which of the two is more likely to have immatures or pupae. Yes, it does appear that you are comparing the first listed against the second listed, but it never states that. Might it be worth a footnote to “Parameter” along the lines of “Likelihood that the first parameter produced more immatures or pupae than the second parameter.” Also, what about the comparison between <5L and 5-10L. 

19. Lines 416-417: Minor, but the number of immature Ae. aegypti stages if both larvae and pupae were present would be 2. The number of “immatures” is what you are examining. I must admit that this is a minor and the readers will understand, but technically, the answer is 0, 1, or 2 for the number of stages present.

20. Line 422: Is it worth a short sentence here. We found HI and BI between 15 and 42 and between17 and 106 for HI and BI, respectively, depending on season and location.

21. Lines 452-455: Yes, I know that plastic containers contributed the most. However, was it because plastic containers were more attractive or a better habitat, or was it because there were so many more plastic containers. If you look at it slightly differently, percent of type of container that was positive for Ae. aegypti immatures, then the order is reversed, with plastic having only 65%, metal being 73%, and tires have a 82% positive rate for Ae. aegypti. This is almost a contradiction of lines 458-462.

22. Line 464 (also 478 and 536): Again, mosquitoes do not “breed” in tires, their immatures develop there.

23. Line 478: Also, the trade in used tires is important as that is an excellent means of transporting Aedes eggs to a new location.

24. Lines 498-499 and 550: why establish “ZAMEP” if you do not use it again. Either do not establish or use on line 55

25. Line 552: Why establish “IDSR” if it is not used. Please check to see if ALL of the abbreviations established are actually used. If not, then please remove these unnecessary abbreviations.

26. References: Please ensure that these are formatted according to PLoS NTD guidelines. Only the first word or proper nouns in a title should be capitalized. See reference 3 and others.

Reviewer's Responses to Questions

**Key Review Criteria Required for Acceptance?**

**Methods**

-Are the objectives of the study clearly articulated with a clear testable hypothesis stated?

-Is the study design appropriate to address the stated objectives?

-Is the population clearly described and appropriate for the hypothesis being tested?

-Is the sample size sufficient to ensure adequate power to address the hypothesis being tested?

-Were correct statistical analysis used to support conclusions?

-Are there concerns about ethical or regulatory requirements being met?

Reviewer #1: See Summary and General Comments

Reviewer #2: (No Response)

**Results**

-Does the analysis presented match the analysis plan?

-Are the results clearly and completely presented?

-Are the figures (Tables, Images) of sufficient quality for clarity?

Reviewer #1: See Summary and General Comments

Reviewer #2: (No Response)

**Conclusions**

-Are the conclusions supported by the data presented?

-Are the limitations of analysis clearly described?

-Do the authors discuss how these data can be helpful to advance our understanding of the topic under study?

-Is public health relevance addressed?

Reviewer #1: See Summary and General Comments

Reviewer #2: (No Response)

**Editorial and Data Presentation Modifications?**

Reviewer #1: See Summary and General Comments

Reviewer #2: For me, the analyzes are not sufficient to accept the publication of this paper in PNTD.

**Summary and General Comments**

Reviewer #1: This manuscript focuses on an Aedes aegypti larval survey in urban and rural sites in Zanzibar. This study to address Aedes aegypti fills a gap of limited knowledge of this vector species in Zanzibar. The motivation for this study is that Aedes control is needed to minimize impacts of arbovirus outbreaks. Ideally, as recommended in the manuscript, this should be integrated with malaria vector control within an IVM framework recommended by WHO.

Overall, this manuscript is well written, with clear results. Both the Introduction and Discussion are informative. The manuscript deals with an important public health problem. It is a good baseline for the further development of surveillance and control operations in Zanzibar.

Recommendations:

1--Throughout, to be accurate, replace the term “breeding” habitat with larval habitat.

2--Line 380 – what type of vegetation in containers? Please indicate here, before you reveal the presence of algae in the Discussion.

3--Line 386 – be careful of saying borderline significant. It can be misleading. I prefer significant or not.

4--Line 498. Integrating Aedes control with malaria vector control is important. Also note this approach is consistent with IVM guidelines recommended by WHO.

5--Line 518. Limitations. Also include as a limitation, the short duration of the study, thought it covered periods of the wet and dry season. Another limitation is rather short duration of in the field mosquito sampling during both wet and dry seasons.

6--Aedes surveillance is important for any efforts in vector control. It will be very helpful if you give some guidance on how this can be done.

7--References 1, 10, 21 and more. Are there references available? Avoid url’s. The high percentage of url’s instead of references detracts from the quality of the manuscript.

8--Figure 1, it is ok but why are the urban and rural sites so far disconnected? Normally, it is common to work in urban areas and select rural sites just outside the urban areas. Selection bias of sites is a major concern, not just in this study; entomologists typically choose sites where they expect to find the most mosquitoes. Was this the case here? More discussion is needed to provide a better rationale for your site selection.

9- Mosquito identifications. It is ok to rear immatures and identify adults. Was this done even for first and second instars? Please clarify in the methods. It is oftentimes preferable to identify larvae, and let pupae emerge. This should be considered in the future.

10—Length of manuscript. The manuscript is long and should be shortened without loosing content. Suggest shortening 10 to 20%.

Reviewer #2: An interesting study assessing the traditional Stegomyia indices and typology of breeding sites in rural and urban areas in Zanzibar. Paper is well written but some clarification and additional analyses are needed before accepting for publication in PNTD.

Lines 42-43. How the productivity was assessed?

Line 66. Ae. Aegypti please correct it.

Material and Method

Which technic was used to select houses for inspection? Did the selected houses cover all the environments of the city? It would be good to add a map indicating the houses selected. The same houses were prospected during the dry and raining season?

The number of pupae and larvae were counted? What was the rate of transformation of larvae into adults?

Why pupae based indexes were not calculated? It would have been good to count the number of people per house to assess the number of pupae per person.

Results

The difference between indices calculated (BI and HI) was significantly different according to the season or environment (urban vs rural)?

What type of container was mainly found indoor?

Table 6 Tanks instead of “tanks”. 

How authors did to differentiate the pupae of Ae. aegypti to others species?

Lines 427-428. It would be fairer to compare the mean number of immature stages per container or house rather than the overall abundance between rural and urban setting.

Line 459. What could explain the propensity of Ae. aegypti to bred in plastic containers?

Higher indices reported in this study are comparable with the case of other African cities? 

Have such high indices previously coincided with true outbreaks occurring?

PLOS authors have the option to publish the peer review history of their article (what does this mean?). If published, this will include your full peer review and any attached files.

Reviewer #1: No

Reviewer #2: No
---

## [Editor Report · Decision Letter 1]

12 Oct 2020

Dear SALEH,

Thank you very much for submitting the revised version of your manuscript "Epidemic risk of arboviral diseases: determining the habitats, spatial-temporal distribution and abundance of immature Aedes aegypti in the Urban and Rural areas of Zanzibar, Tanzania" for consideration at PLOS Neglected Tropical Diseases. This version is much improved. Thanks for being so clear in your responses to me and the reviewers. We are likely to accept this manuscript for publication, providing that you modify the manuscript according to my recommendations below.

1. I had a hard time correlating the line number indicated in the response letter and the line number in the revised manuscript. For example, in the response letter it said, “The footnote inserted on Table 7 and 8 as suggested – Lines 476 – 477 and 481-482.” However, those footnotes were on lines 377-378 and 482-483. Similarly, in response to the comment about Dr. Mohammed’s work, the authors responded, “…which we refer in the revised manuscript (Ref 20), - Lines 102 - 103.” However, that was inserted on lines 99-100. Please check to ensure that the line numbers mentioned are accurate for the final, revised, clean copy.

2. Minor, but is it one space or two spaces between sentences? Either is fine, but be consistent. I also noted some extra spaces between words. You might want to do a search for “space space” (obviously not the words) and remove the extra ones. 

3. Line 43 and others: Again, as indicted earlier, the number of immature “stages” observed in a given sample is only 0, 1, or 2. You are not reporting the number of “stages,” but the number of immatures. I would remove “stages” throughout much of the manuscript. Note, “immature stages or pupae” is confusing. I believe that what you are looking at is all immatures (i.e., larvae and pupae) or pupae only. I would consider replacing the phrase, “immature stages or pupae” with “all immatures or pupae only” throughout the manuscript. Note that on line 228 you clearly indicated that “immature stages” indicates both larvae and pupae, but it still bothers me as you are not trying to measure the number of “stages,” but rather the number of “all immatures” or just the number of pupae. When you are just referring to immatures, i.e., line lines 37, 202, 214, 219, 228, 235, 273, etc. (i.e., where it does NOT say immature stage or pupae) then the use of “stages” is appropriate. However, the use of “stages” on lines 43, Table 3, 308, 317, Table 5, 336, Table 6, 398, 424,425, and 426 is not appropriate.

4. Lines 255-257: Yes, the rate was much higher, but was it significantly different? Actually, it is at a p-value <0.0001, so you might want to add that the difference was highly significantly different. Same comment for wet versus dry.

5. Lines 301-306. Similar comment. Yes, more containers were identified indoors than outdoors, but was this difference significant? 

6. Line 321, Table 5: I just rechecked this table, and there are a number of problems. 

a. Did you really look for “immature stages and pupae?” Wouldn’t looking for immature stages include looking for pupae? I would change the title to either “…Ae. aegypti immatures in Zanzibar” or possibly to “…immature Ae. aegypti in Zanzibar.” 

b. Unfortunately, the way the spacing is done makes it difficult to see which groups are being compared. For example, having “Organic Matter” cover two lines makes it look like Organic matter “Yes” is part of the Vegetation group. Similarly, by having Water volume taking up two lines, it makes Water volume “5L-20L” and “>20 L” look like they are being compared with Sun exposure “<½ day” rather than with Water volume “<5L.” Thus, because there are a lot of columns, it might be better to use a landscape (horizontal) layout for this table. You might also change “Exposed half a day or less” to “Exposed < half day” or even “Exposed <½ day” and “Exposed >½ day” to shorten that area.

c. The layout is also a little confusing. The first column of “Containers positive” is for all immature states, while the second one is for only those positive for pupae. But this is not clear. Could you make it “Containers positive for immatures” and “Containers positive for pupae?” Also, should it be “No. of immature stages (%2)” or “No. of immatures (%2)?” Remember, there are two stages, larvae and pupae, so the real number of stages is two, which is different than the number of immatures present. Delete “stages” and use “No. of immatures (%2).”

d. I just rechecked the statistics, and it appears that 156/433 (36%) is highly significantly different than 152/660 (23%) by a Fisher’s exact test (p < 0.0001) for organic matter. Similarly, <5L is NOT significantly different than >20L (Fisher’s exact test, p = 0.282). When I originally wrote this comment, I had not checked the statistics, so I was just illustrating how to write it. You need to go back through and actually check the statistics for each of these, and then in the footnotes indicated how you did it and what you mean by “statistically different.” For example, the footnote could be, “The categories followed by different letters (a, b) within a given parameter were significantly different (Fisher’s exact test, p < 0.05) and those followed by the same letter (a, a, or b, b) were not significantly different.” Remember, all data needs to be presented in the past tense.

7. Line 338, Table 6: To me, another important question would have been what percentage of the various types of containers were actually positive for immatures or pupae. If I calculated it properly, 65% (344/529) of the plastic containers were positive, 73% (172/235) of the metal containers were positive, 82% (134/164) of the tires were positive. It might be nice to add this, possibly in parentheses after the number positive, so for Plastic, under positive containers, it would be “344 (65%)” with a footnote on containers that said, “Number of containers positive (percent of that type of container positive for Ae. aegypti)

8. Lines 357-360: I ran both a Chi-square and a Fisher’s exact test on the <5L vs >20 liter, and the p-values were >0.28 as indicated above. Please check your statistics.

9. Line 447: Minor, but “as it present an…” should be “as it presents an..”

10. Line 465. Yes, Anopheles was established in the introduction, but that does not establish “An.” The first time you mention an Anopheles species, you need to write out Anopheles, or in this case, “Anopheles gambiae.”

11. Line 563, Ref 11: Dengue Virus Type 1 should be “Dengue virus type 1…” as only the first word and proper nouns should be capitalized.

12. Supplemental table: I just wanted to confirm that out of the 13,677 Aedes larvae captured and identified, 13,671 were Ae. aegypti. That is very impressive

Sincerely,

Michael J Turell, Ph.D.

Associate Editor

Amy Morrison

Deputy Editor
---

## [Editor Report · Decision Letter 2]

23 Oct 2020

Dear SALEH,

Thank you very much for submitting your revised version of the manuscript "Epidemic risk of arboviral diseases: determining the habitats, spatial-temporal distribution and abundance of immature Aedes aegypti in the Urban and Rural areas of Zanzibar, Tanzania" for consideration at PLOS Neglected Tropical Diseases. I appreciate your careful responses and believe that the manuscript is nearly ready to accept. I have a few comments below that need to be addressed. 

1. Lines 272-277: You wrote that the positivity rate for containers for immatures was significantly higher in the wet as compared to the dry season, but never mentioned that the rate was or was not significantly different for pupae (it is obviously not significant, but you never said so. Note, don’t merely write “These rates were not significantly different p > 0.05.” Calculate and indicate the actual p-value. Note, if the value was 0.06, it would not be significant, but given that the immatures were significant, this would indicate that the difference in pupae was likely to be real as 0.05 is an arbitrary cut off value. However, in the current case, I believe that the p-value is 0.94, by a Fisher’s exact test and clearly not significant. Also, it is interesting to note that the number positive for pupae in the wet season, 236, is again nearly three times greater than the number positive for pupae in the dry season, 82. This difference in number positive is not due to there being a higher positivity rate in the wet season, but due to there being nearly three times as many containers examined in the wet season. Thus, the initial sentence, “More than three times greater numbers of containers were positive for immature Ae. aegypti in the wet (N = 543/814, 66.7%) compared with dry (N = 165/279, 59.1%) seasons…” is a little misleading. The huge difference in numbers positive is mostly due to a much larger sample size in the wet season.

2. Line 300: As mentioned above, rather than merely write “(p > 0.05),” please put the actual p-value here any anywhere else only “> 0.05” is used.

3. Lines 300-302: Were these differences in container types significant?

4. Table 5:

a. Yes, I understand the difference and advantages of both GLMM and either a Fisher’s exact test or a Chi-square test, but I am still bothered by the apparent significant difference in Table 5 of “<5L” and both “5L-20L” and “>20L.” As we both noticed, these are not even close to significant by a univariate test. Are containers <5L really significantly more likely to contain immatures than larger containers or not? Looking at Table 5, this difference is “significant” only for the all immatures, but even close to significant for the pupae, yet for the <5L containers, the percent of the immatures positive (59%) was identical to the percent of the of the pupae positive (59%). I think that the “significant” difference by GLMM was due to chance. However, I will leave how to present the statistics up to you.

b. Minor, but in footnotes 2 and 3 for Table 5, it says “immatures or pupae.” Because pupae are immatures, this is not correct. How about, “…immatures for each category…” and “…immatures within each category?” Note, in footnote 1 you wrote that “immatures” means larvae plus pupae. 

Sincerely,

Michael J Turell, Ph.D.

Associate Editor

Amy Morrison

Deputy Editor
---

## [Editor Report · Decision Letter 3]

4 Nov 2020

Dear SALEH,

We are pleased to inform you that your manuscript 'Epidemic risk of arboviral diseases: determining the habitats, spatial-temporal distribution and abundance of immature Aedes aegypti in the Urban and Rural areas of Zanzibar, Tanzania' has been provisionally accepted for publication in PLOS Neglected Tropical Diseases. I hope you agree that the current version of the manuscript is significantly improved.

Best regards,

Michael J Turell, Ph.D.

Associate Editor

Amy Morrison

Deputy Editor

---

## [Editor Report · Acceptance letter]

23 Nov 2020

Dear SALEH,

We are delighted to inform you that your manuscript, "Epidemic risk of arboviral diseases: determining the habitats, spatial-temporal distribution and abundance of immature Aedes aegypti in the Urban and Rural areas of Zanzibar, Tanzania," has been formally accepted for publication in PLOS Neglected Tropical Diseases.

Best regards,

Shaden Kamhawi

co-Editor-in-Chief

Paul Brindley

co-Editor-in-Chief
